# An evolutionarily young defense metabolite influences the root growth of plants via the ancient TOR signaling pathway

Frederikke Gro Malinovsky[1], Marie-Louise F Thomsen[1], Sebastian J Nintemann[1], Lea Møller Jagd[1], Baptiste Bourgine[1†], Meike Burow[1], Daniel J Kliebenstein[1,2]*

[1]DynaMo Center, Copenhagen Plant Science Center, Department of Plant and Environmental Sciences, University of Copenhagen, Copenhagen, Denmark; [2]Department of Plant Sciences, University of California, Davis, Davis, United States

**Abstract** To optimize fitness a plant should monitor its metabolism to appropriately control growth and defense. Primary metabolism can be measured by the universally conserved TOR (Target of Rapamycin) pathway to balance growth and development with the available energy and nutrients. Recent work suggests that plants may measure defense metabolites to potentially provide a strategy ensuring fast reallocation of resources to coordinate plant growth and defense. There is little understanding of mechanisms enabling defense metabolite signaling. To identify mechanisms of defense metabolite signaling, we used glucosinolates, an important class of plant defense metabolites. We report novel signaling properties specific to one distinct glucosinolate, 3-hydroxypropylglucosinolate across plants and fungi. This defense metabolite, or derived compounds, reversibly inhibits root growth and development. 3-hydroxypropylglucosinolate signaling functions via genes in the ancient TOR pathway. If this event is not unique, this raises the possibility that other evolutionarily new plant metabolites may link to ancient signaling pathways.
DOI: https://doi.org/10.7554/eLife.29353.001

*For correspondence:
kliebenstein@ucdavis.edu

Present address: †Department of Plant Molecular Biology, University of Lausanne, Lausanne, Switzerland

## Introduction

Herbivory, pathogen attacks and weather fluctuations are just some of the factors that constantly fluctuate within a plants environment. To optimize fitness under this wide range of conditions, plants utilize numerous internal and external signals and associated signaling networks to plastically control metabolism and development (*Henriques et al., 2014*; *Smith and Stitt, 2007*; *Rexin et al., 2015*). This metabolic and developmental plasticity begins at seed germination, where early seedling growth is maintained by heterotrophic metabolism relying solely on nutrients and energy stored in the seed including the embryo. Upon reaching light, the seedling transitions to autotrophy by shifting metabolism to initiate photosynthesis and alters development to maximize photosynthetic capacity (*Rexin et al., 2015*; *Galili et al., 2014*). Until light is available, it is vital for the plant to prioritize usage from the maternal energy pool, to ensure the shoot will breach the soil before resources are depleted. Because the time to obtaining light is unpredictable, seedlings that had the ability to measure and accordingly adjust their own metabolism would likely enjoy a selective advantage. In this model, energy availability would an essential cue controlling growth throughout a plant's life and not solely at early life-stages. On a nearly continuous basis, photo-assimilates, such as glucose and sucrose, are monitored and their internal levels used to determine the growth potential by partitioning just the right amount of sugars between immediate use and storage (*Smith and Stitt, 2007*).

**eLife digest** Plants, like all organisms, must invest their resources carefully. Growing new roots or shoots may allow a plant to better exploit its environment. But a plant should never leave itself vulnerable to disease. As such, there must be a balance between allocating resources to growth or to defense.

Brassicas like cabbage, Brussels sprouts and wasabi use unique compounds called glucosinolates to protect themselves against pests and disease-causing microbes. These same compounds give these vegetables their distinctive flavors, and they are the source of many of the health benefits linked to eating these vegetables. Yet it was not known if glucosinolates could also affect a plant's growth and development.

Malinovsky et al. tested a number of purified glucosinolates with the model plant *Arabidopsis thaliana,* and found that one (called 3-hydroxypropylglucosinolate) caused the plants to grow with stunted roots. When 10 other species of plant were grown with this glucosinolate, almost all had shorter-than-normal roots. The effect was not limited to plants; baker's yeast also grew less when its liquid media contained the plant-derived compound.

The reason glucosinolates can protect plants against insect pests, provide us with health benefits, and widely inhibit growth is most likely because they have evolved to interact with proteins that are found in many different organisms.Indeed, through experiments with mutant *Arabidopsis* plants, Malinovsky et al. revealed that their glucosinolate influences the TOR complex. This complex of proteins works in an ancient and widespread signaling pathway that balances growth and development with the available energy and nutrients in organisms ranging from humans to yeast to plants.

The TOR complex plays such a vital role in living cells that problems with this complex have been linked to diseases such as cancer and heart disease. Importantly, the chemical structure of this glucosinolate is unlike other compounds that have already been tested against the TOR complex. As such, it is possible that this glucosinolate might lead to new drugs for a range of human diseases. Further, as this compound affects plant growth, it could also act as a starting point for new herbicides.

Together these findings show how studying molecules made in model organisms and understanding how they function can lead to the identification of new compounds and targets with an unexpectedly wide range of potential uses.

DOI: https://doi.org/10.7554/eLife.29353.002

Illustrating the key nature of metabolite measurement within plants is that glucose, is measured by two separate kinase systems that are oppositely repressed and activated to determine the potential growth capacity, SnRKs1 (sucrose non-fermenting 1 (SNF1)-related protein kinases 1) and the Target of Rapamycin (TOR) kinase (*Sheen, 2014*). SnRKs1s are evolutionarily conserved kinases that are activated when sugars are limiting (*Lastdrager et al., 2014*). *Arabidopsis thaliana* (Arabidopsis) has two catalytic SnRK1-subunits, KIN10 and KIN11 (SNF kinase homolog 10 and 11), that activate vast transcriptional responses to repress energy-consuming processes and promote catabolism (*Baena-González, 2010*; *Crozet et al., 2014*; *Baena-González et al., 2007*). This leads to enhanced survival during periods of energy starvation. Oppositely, the TOR kinase is a central developmental regulator, whose sugar-dependent activity controls a myriad of developmental processes including cell growth, cell-cycle, and cell-wall processes. The TOR pathway functions to modulate growth and metabolism by altering transcription, translation, primary and secondary metabolism, as well as autophagy (*Sheen, 2014*; *Sablowski et al., 2014*). The TOR kinase primarily functions in meristematic regions where it promotes meristem proliferation. Within these cell types, TOR measures the sugar content and if the tissue is low in sugar, TOR halts growth, even overruling hormone signals that would otherwise stimulate growth (*Xiong et al., 2013*). In plants, TOR functions within a conserved complex that includes RAPTOR (regulatory-associated protein of TOR) and LST8 (lethal with sec-13 protein 8) (*Henriques et al., 2014*). RAPTOR likely functions as an essential substrate-recruiting scaffold enabling TOR substrate phosphorylation (*Rexin et al., 2015*), and LST8 is a seven WD40 repeats protein with unclear function (*Moreau et al., 2012*). TOR complex (TORC) activity is

positively linked with growth (*Rexin et al., 2015*) as mutants in any component lead to qualitative or quantitative defects in growth and development and even embryo arrest in strong loss-of-function alleles (*Menand et al., 2002*; *Deprost et al., 2005*). Although the energy sensory kinases KIN10/11 and TOR sense opposite energy levels, they govern partially overlapping transcriptional networks, which are intimately connected to glucose-derived energy and metabolite signaling (*Sheen, 2014*; *Baena-González et al., 2007*). Having two systems to independently sense sugar shows the importance of measuring internal metabolism. A key pathway controlled by TOR in all eukaryotes is autophagy (*Liu and Bassham, 2010*; *Shibutani and Yoshimori, 2014*). In non-stressed conditions, continuous autophagy allows the removal of unwanted cell components like damaged, aggregated or misfolded proteins by vacuolar/lysosomal degradation (*Inoue et al., 2006*). Under low energy conditions, TORC inhibition leads to an induction of autophagy to free up energy and building blocks, through degradation of cytosolic macromolecules and organelles (*Shibutani and Yoshimori, 2014*). Autophagy-mediated degradation is facilitated by formation of autophagosomes; double membrane structures that enclose cytoplasmic cargo, and delivers it to the vacuole (*Shibutani and Yoshimori, 2014*; *Feng et al., 2014*; *Le Bars et al., 2014*; *Zhuang et al., 2016*).

In nature, plant plasticity is not only limited to responding to the internal energy status, but to an array of external environmental inputs. The multitude of abiotic and biotic factors that plants continuously face often require choices between contradictory responses, that requires integrating numerous signals across an array of regulatory levels to create the proper answer. For example; plant defense against biotic organisms requires coordination of metabolic flux to defense and development while in continuous interaction with another organisms and the potential for interaction with other organisms. A proper defense response is vital for the plant as a metabolic defense response to one organism can impart an ecological cost by making the plant more sensitive to a different organism (*Züst et al., 2012*). Therefore, a plant must choose the most appropriate defense response for each situation to optimize its fitness and properly coordinate its defense response with growth and development. A key defense mechanism intricately coordinated across development is the synthesis of specific bioactive metabolites that are often produced in discrete tissues at specific times. A current model is that developmental decisions hierarchically regulate defense metabolism with little to no feed-back from defense metabolism to development. However, work on the glucosinolate and phenolic pathways is beginning to suggest that defense metabolites can equally modulate development (*Katz et al., 2015*; *Bonawitz et al., 2012*; *Bonawitz et al., 2014*; *Kim et al., 2014*; *Francisco et al., 2016a*; *Francisco et al., 2016b*), thus suggesting that development and defense metabolism can directly cross-talk.

To asses if and how defense metabolites can signal developmental changes, we chose to investigate the glucosinolate (GSL) defense metabolites. The evolution of the core of GSL biosynthesis is relatively young, and specifically modified GSL structures are even more recent (*Sønderby et al., 2010*). There are >120 known GSL structures limited to plants from the *Brassicales* order and some *Euphorbiaceae* family members, with Arabidopsis containing at least 40 structures (*Sønderby et al., 2010*). GSLs are amino acid derived defense metabolites that, after conversion to an array of bioactive compounds, provide resistance against a broad suite of biotic attackers (*Sønderby et al., 2010*; *Lambrix et al., 2001*; *Kliebenstein et al., 2002*). GSLs not only exhibit a wide structural diversity, but their composition varies depending on environmental stimuli, developmental stage and even across tissues. This, combined with the information-rich side chain, makes GSLs not only an adaptable defense system, but also prime candidates for having distinctive signaling functions. Previous work has suggested that there may be multiple signaling roles within the GSLs (*Francisco et al., 2016a*; *Francisco et al., 2016b*; *Clay et al., 2009*; *Khokon et al., 2011*; *Bednarek et al., 2009*). Tryptophan-derived indole GSLs or their breakdown products alter defense responses to non-host pathogens illustrated by biosynthetic mutants devoid of indole GSLs being unable to deposit PAMP-induced callose in cell walls via an unknown signal and pathway (*Clay et al., 2009*; *Bednarek et al., 2009*). Similarly, indole GSL activation products have the ability to directly alter auxin perception by interacting with the TIR1 auxin receptor (*Katz et al., 2015*). In contrast to indole GSLs, mutants in aliphatic GSL accumulation alter flowering time and circadian clock oscillations (*Kerwin et al., 2011*; *Jensen et al., 2015*). The aliphatic GSL activation product, allyl isothiocyanate can induce stomatal closure but it is unknown if this is specific to allylGSL or a broader GSL property (*Khokon et al., 2011*; *Boller and Felix, 2009*). AllylGSL (other names are 2-propenylGSL and sinigrin) can also alter plant biomass and metabolism in Arabidopsis (*Francisco et al., 2016a*; *Francisco et al., 2016b*).

While these studies have provided hints that the GSL may have signaling potential, there is little understanding of the underlying mechanism or the structural specificity of the signal. To explore whether built-in signaling properties are a common attribute of GSLs, we screened for altered plant growth and development in the presence of specific purified aliphatic GSLs. In particular, we were interested in identifying candidate signals whose activity could ensure fast repartitioning of resources between development and defense. Here we present a novel signaling capacity specific to the aliphatic 3-hydroxypropylglucosinolate (3OHPGSL). Our results suggest that 3OHPGSL signaling involves the universally conserved TOR pathway for growth and development, as mutants in TORC and autophagy pathways alter responsiveness to 3OHPGSL application.

## Results

### 3ohpgsl inhibits root growth in arabidopsis

We reasoned that if a GSL can prompt changes in plant growth it is an indication of an inherent signaling capacity. Using purified compounds, we screened for endogenous signaling properties among short-chain methionine-derived aliphatic GSLs by testing their ability to induce visual phenotypic responses in Arabidopsis seedlings. We found that 3OHPGSL causes root meristem inhibition, at concentrations down to 1 µM (*Figure 1A*). The observed response is concentration-dependent (*Figure 1A–B*). All Arabidopsis accessions accumulate 3OHPGSL in the seeds, and this pool is maintained at early seedling stages (*Chan et al., 2011*; *Petersen et al., 2002*; *Brown et al., 2003*). Previous research has shown that GSLs in the seed are primarily deposited in the embryo, accumulating to about 3µmol/g suggesting that we are working with concentrations within the endogenous physiological range (*Fang et al., 2012*; *Kliebenstein et al., 2007*).

We tested how exogenous 3OHPGSL exposure to the roots alters 3OHPGSL accumulation within the shoot, and how this compares to endogenously synthesized 3OHPGSL levels. We grew the Col-0 reference accession and the *myb28-1 myb29-1* mutant that is devoid of endogenous aliphatic GSLs in the presence and absence of exogenous 3OHPGSL. At day 10, the foliar 3OHPGSL levels were analyzed. Col-0 without treatment had average foliar levels of 3OHPGSL of 3.2 µmol/g and grown on media containing 5 µM, 3OHPGSL contributed an additional 2.2 µmol/g raising the total 3OHPGSL to 5.4 µmol/g (*Figure 1C*). The *myb28-1 myb29-1* mutant had no measurable 3OHPGSL on the control plates and accumulated ~1.4µmol/g upon treatment (*Figure 1D*). In agreement with the lower foliar 3OHPGSL accumulation in the *myb28-1 myb29-1* mutant background, this double mutant had a lower root growth response to exogenous 3OHPGSL (*Figure 1—figure supplement 1*). Importantly, this confirms that the level of 3OHPGSL application is within the physiological range.

We then tested if 3OHPGSL or potential activation products inhibit root growth because of cell death or toxicity. The first evidence against toxicity came from the observation that even during prolonged exposures, up to 14 days of length, Col-0 seedlings continued being vital and green (*Figure 1E*). If there was toxicity the seedlings would be expected to senesce and die. We next tested if the strong root growth inhibition by 50 µM 3OHPGSL is reversible. Importantly, root inhibition is reversible, as the 3OHPGSL-mediated root stunting could be switched on and off by transfer between control media and media containing 3OHPGSL (*Figure 1E*). Based on the toxicity and GSL assays, we conclude that the 3OHPGSL treatments are at reasonable levels compared to normal Arabidopsis physiology, and that the phenotypic responses we observed were not caused by flooding the system with 3OHPGSL or toxicity.

### Root inhibition is specific to 3OHPGSL

To evaluate whether 3OHPGSL mediated root inhibition is a general GSL effect or if it is structurally specific to 3OHPGSL, we tested if aliphatic GSLs with similar side-chain lengths, but different chain modifications, would induce similar root growth effects. First, we assessed 3-methylsulfinyl-propyl (3MSP) GSL, the precursor of 3OHPGSL, and the alkenyl-modified three carbon glucosinolate allyl (*Figure 2A*). In contrast to 3OHPGSL, neither of these structurally related GSLs possessed similar root-inhibiting activities within the tested concentration range (*Figure 2B–D*). We also analyzed the potential root inhibition for the one carbon longer C4-GSLs 4-methylsulfinylbutyl (4MSB) and but-3-enyl (*Figure 2E*). Neither of these compounds could inhibit root growth at the tested concentrations (*Figure 2F–H*). There is no viable commercial, synthetic or natural source for the 4-hydroxybutyl GSL

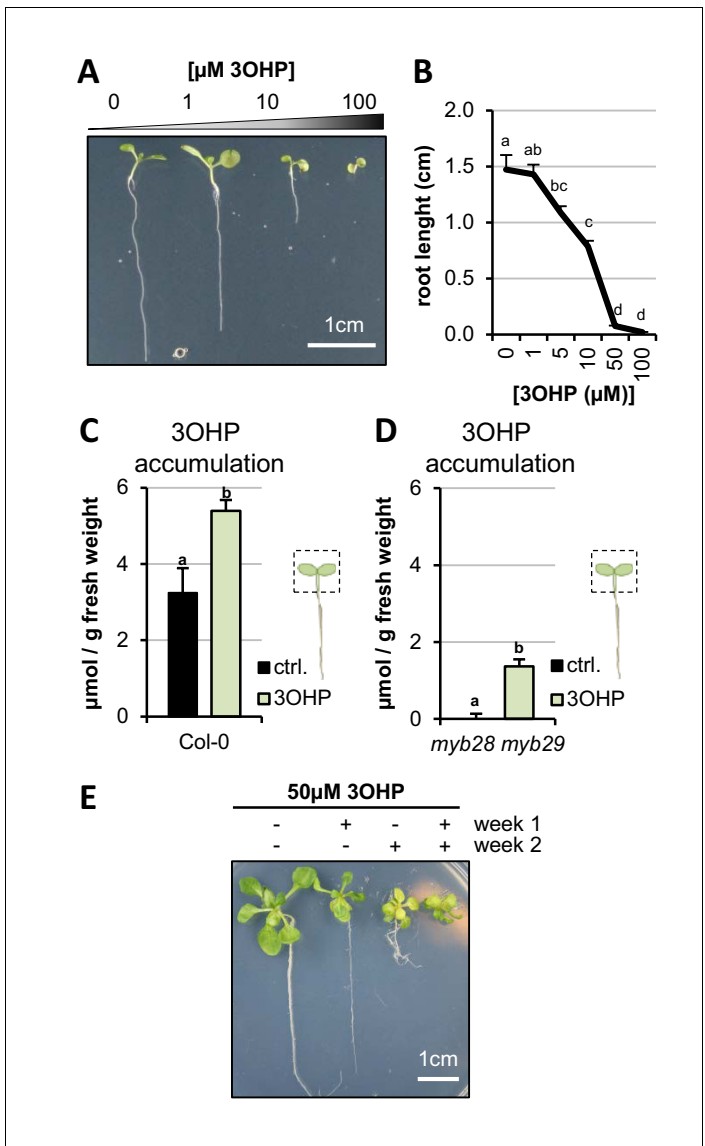

**Figure 1.** 3OHP reversibly inhibits root growth. (A) 7-d-old seedlings grown on MS medium supplemented with a concentration gradient of 3OHP. (B) Quantification of root lengths of 7-d-old. Results are averages ± SE (n = 3–7; p<0.001). (C) Accumulation of 3OHP in shoots/areal tissue of 10-d-old Col-0 wildtype seedlings grown on MS medium supplemented with 5 μM 3OHP. Results are least squared means ± SE over three independent experimental replicates with each experiment having an average eleven replicates of each condition (n = 31–33; ANOVA P$_{Treat}$ < 0.001). (D) Accumulation of 3OHP in shoots of 10-d-old *myb28 myb29* seedlings (aliphatic GSL-free) grown on MS medium supplemented with 5 μM 3OHP. Results are least squared means ± SE over two independent experimental replicates with each experiment having an average of four independent biological replicates of each condition (n = 8–14; ANOVA P$_{Treat}$ < 0.001). (E) 14-d-old seedlings grown for 1 week with or without 3OHP as indicated. After one week of development, the plants were moved to the respective conditions showed in week 2.

DOI: https://doi.org/10.7554/eLife.29353.003

The following figure supplement is available for figure 1:

**Figure supplement 1.** Root inhibition is affected by endogenous GSL levels.

DOI: https://doi.org/10.7554/eLife.29353.004

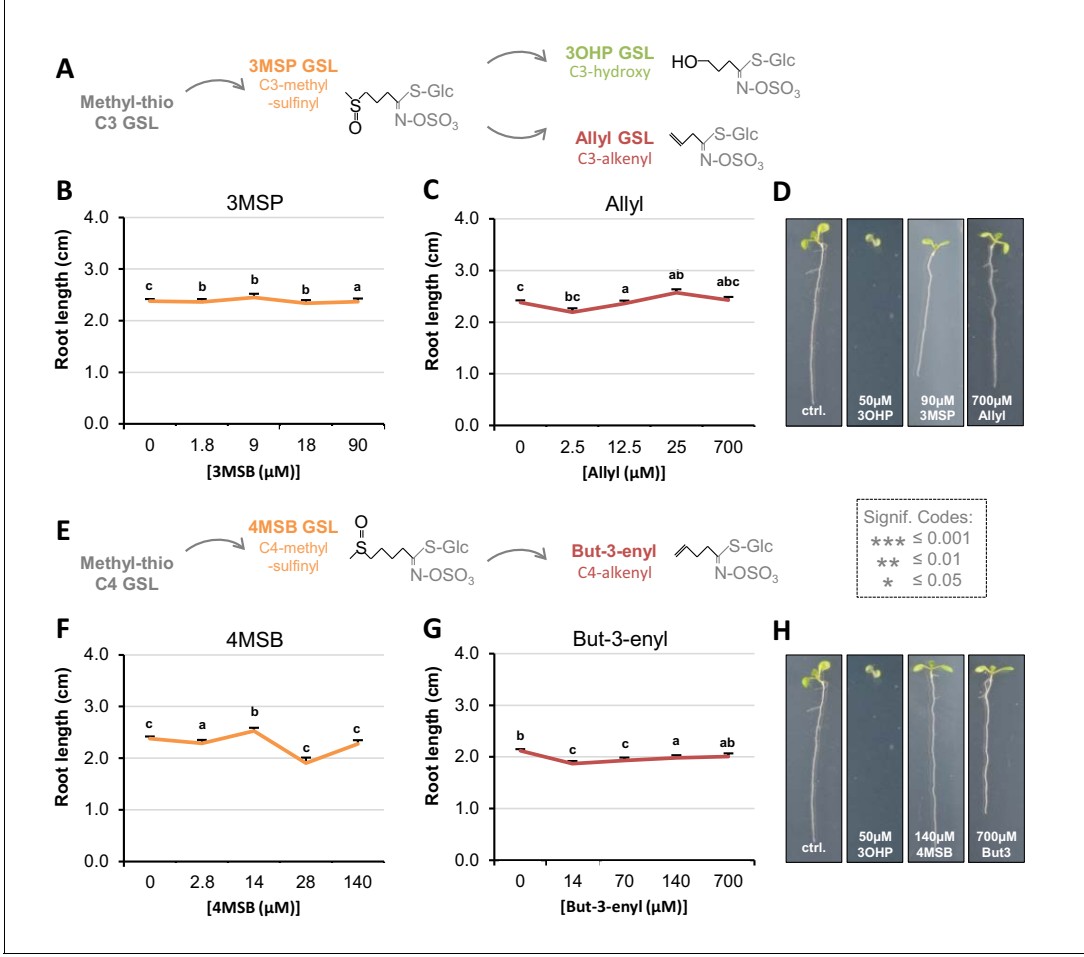

**Figure 2.** Root growth is not inhibited by all aliphatic GSLs. (A) The aliphatic glucosinolate biosynthetic pathway, from the C3 3-methyl-sulphinyl-propyl (3MSP) to the secondary modified 3-hydroxyl-propyl (3OHP) and 2-propenyl (allyl/sinigrin). (B–C) Root lengths of 7-d-old Col-0 wildtype seedlings grown on MS medium supplemented with a concentration gradient of the indicated aliphatic C3-GSL. The left most point in each plot shows the root length grown in the absence of the specific GSL treatment. Results are least squared means ±SE over four independent experimental replicates with each experiment having an average of 21 replicates per condition ($n_{3MSP}$=59–153; $n_{Allyl}$ = 52–153). Significance was determined via two-way ANOVA combining all experiments. (D) 7-d-old seedlings grown on MS medium with or without 50 μM of the indicated GSL. (E) The aliphatic glucosinolate biosynthetic pathway from the C4 4-methyl-sulphinyl-butyl (4MSB) to But-3-enyl. (F–G) Root lengths of 7-d-old Col-0 wildtype seedlings grown on MS medium supplemented with a concentration gradient of the indicated aliphatic C4-GSL. The left most point in each plot shows the root length grown in the absence of the specific GSL treatment. Least squared means ±SE over four independent experimental replicates with each experiment having an average of 22 replicates condition ($n_{4MSB}$=38–153; $n_{But-3-enyl}$=68–164). Significance was determined via two-way ANOVA combining all experiments. (H) 7-d-old seedlings grown on MS medium with or without 50 μM of the indicated GSL.

DOI: https://doi.org/10.7554/eLife.29353.005

The following figure supplement is available for figure 2:

**Figure supplement 1.** Root lengths of 7 DAG Col-0 WT grown on MS media supplemented with 50 μM of the indicated GSL.
DOI: https://doi.org/10.7554/eLife.29353.006

which prevented us from testing this compound. To test if the presence of a hydroxyl is essential for this response, we tested if either the R or S form of 2-hydroxybut-3-enylGSL had similar effects. This showed that neither enantiomer was similar to 3OHPGSL and that both actually stimulated root growth (*Figure 2—figure supplement 1*) The fact that only 3OHPGSL inhibits root elongation suggests that the core GSL structure (comprised of a sulfate and thioglucose) does not cause the effect. Importantly, this indicates that 3OHPGSL root inhibition is not a generic result of providing extra sulfur or glucose from the GSL core structure to the plant, as these compounds would be equally contributed by the other GSLs. Furthermore, the results confirm that there is no general toxic activity when applying GSLs to Arabidopsis. This evidence argues that the 3OHPGSL root inhibition effect

links to the specific 3OHP side chain structure, indicating the presence of a specific molecular target mediating the root inhibition response.

## 3OHPGSL responsiveness is wider spread in the plant kingdom than GSL biosynthesis

The evolution of the GSL defense system is a relatively young phylogenetic event that occurred within the last ~92 Ma and is largely limited to the Brassicales order (*Edger et al., 2015*). The aliphatic GSL pathway is younger still (~60 Ma) and is limited to the Brassicaceae family with the enzyme required for 3OHPGSL production, AOP3, being limited to *Arabidopsis thaliana* and *Arabidopsis lyrata* within the Arabidopsis lineage (*Edger et al., 2015*; *Kliebenstein et al., 2001a*). However, 3OHPGSL is also found in the vegetative tissue of the close relative *Olimarabidopsis pumila* (dwarf rocket) (*Windsor et al., 2005*), and in seeds of more distant Brassicaceae family members such as the hawkweed-leaved treacle mustard (*Erysimum hieracifolium*), virginia stock (*Malcolmia maritima*), shepherd's cress (*Teesdalia nudicaulis*), and alpine pennycress (*Thlaspi alpestre*) (*Daxenbichler et al., 1980*; *Daxenbichler et al., 1991*). These species are evolutionarily isolated from each other, suggesting that they may have independently evolved the ability to make 3OHPGSL (*Kliebenstein et al., 2001a*; *Windsor et al., 2005*; *Cni, 2016*). As such, 3OHPGSL is an evolutionarily very young compound and we wanted to determine if the molecular pathway affected by 3OHPGSL is equally young, or whether 3OHPGSL affects an evolutionarily older, more conserved pathway.

First we tested for 3OHPGSL responsiveness in plant species belonging to the GSL-producing Brassicales order (*Figure 3A*). We found that four of the five tested Brassicales species responded to 5 μM 3OHPGSL with root growth inhibition regardless of their ability to synthesize 3OHPGSL (*Figure 3B–F*). This suggests that responsiveness to 3OHPGSL application does not link to the ability to make 3OHPGSL. We expanded the survey by including plants within the eudicot lineage that do not have the biosynthetic capacity to produce any GSLs (*Figure 3G*) and found that 5 μM 3OHPGSL can inhibit root growth in several of the non-Brassicales species tested (*Figure 3H–L*). The ability of 3OHPGSL to alter growth extended to *Saccharomyces cerevisae* where 3OHPGSL led to slower log phase growth than the untreated control (*Figure 3—figure supplement 1*). Allyl GSL in the media had no effect on *S. cerevisae* growth showing that this was a 3OHPGSL mediated process (*Figure 3—figure supplement 1*). The observation that 3OHPGSL responsiveness is evolutionarily older than the ability to synthesize 3OHPGSL suggests that the molecular target of

## 3OHP reduces root meristem and elongation zone sizes

We hypothesized that 3OHPGSL application may alter root cellular development to create the altered root elongation phenotype. A reduction in root growth can be caused by inadequate cell division in the root meristematic zone or by limited cell elongation in the elongation zones (*Figure 4A*) (*Henriques et al., 2014*). To investigate how 3OHPGSL affects the root cellular morphology, we used confocal microscopy of 4-d-old Arabidopsis seedlings grown vertically with or without 10 μM 3OHPGSL. We used propidium iodide stain to visualize the cell walls of individual cells, manually counted the meristematic cells, and measured the distance to the point of first root hair emergence. Root meristems of 3OHPGSL treated seedlings were significantly reduced in cell number compared to untreated controls (*Figure 4B–C*). Moreover, we also observed a premature initiation of the differentiation zone, as the first root hairs were closer to the root tip upon 3OHPGSL treatment (*Figure 4D–E*). In addition, we saw bulging and branching of the root hairs in 3OHPGSL treated roots (*Figure 4F*). There was no morphological evidence of cell death in any root supporting the argument that 3OHPGSL is not a toxin. These results indicate that 3OHPGSL leads to root growth inhibition by reducing the size of the meristematic zone within the developing Arabidopsis root.

## TORC-associated mutants alter 3OHPGSL responsiveness

The observed response to 3OHPGSL suggests that the target of this compound is evolutionarily conserved and alters root growth but does not affect the patterning of the root meristem. This indicates that key root development genes like SHR and SCR are not the targets as they affect meristem patterning (*Sabatini et al., 2003*; *Hao and Cui, 2012*). Mutants in GSL biosynthetic genes can lead to

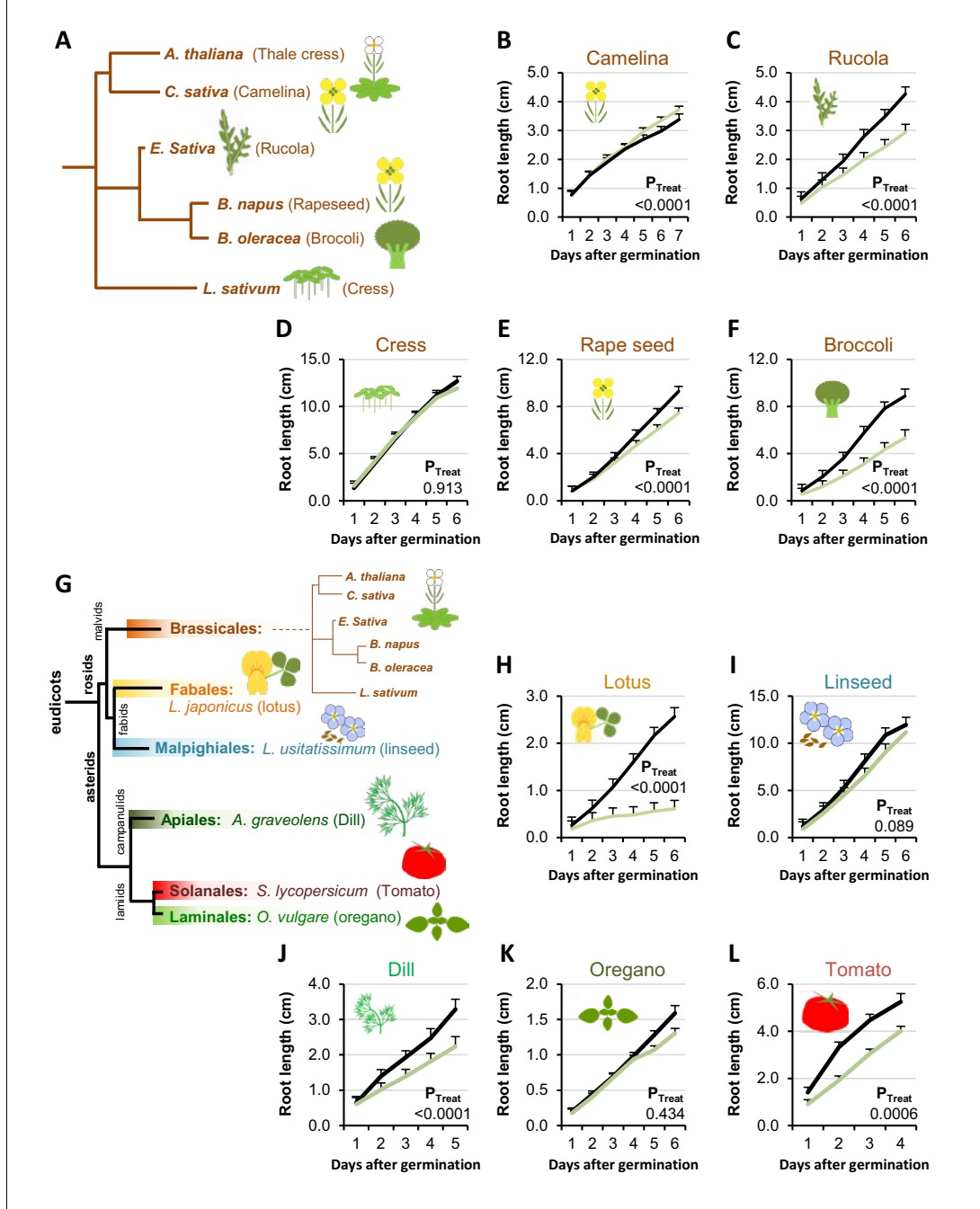

**Figure 3.** Conservation of 3OHP responsiveness suggests a evolutionally conserved target. (**A**) Stylized phylogeny showing the phylogenetic relationship of the selected plants from the Brassicales family, branch lengths are not drawn to scale. (**B–F**) plants from the Brassicales family, grown on MS medium supplemented with or without 5 μM 3OHP. (**G**) Stylized phylogeny showing the phylogenetic relationship of all the selected crop and model plants, branch lengths are not drawn to scale. (**H–L**) Root growth of plants from diverse eudicot lineages, grown on MS medium supplemented with or without 5 μM 3OHP. Results are least squared means ± SE for each species using the following number of experiments with the given biological replication. Camelina three independent experimental replicates ($n_{ctrl}$ = 8 and $n_{3OHP}$=12). Rucola three independent experimental replicates ($n_{ctrl}$ = 17 and $n_{3OHP}$=17. Cress; three independent experimental replicates ($n_{ctrl}$ = 19 and $n_{3OHP}$=18). Rape; seed four independent experimental replicates ($n_{ctrl}$ = 14 and $n_{3OHP}$=13). Broccoli; three independent experimental replicates ($n_{ctrl}$ = 10 and $n_{3OHP}$=13). Lotus; three independent experimental replicates ($n_{ctrl}$ = 10 and $n_{3OHP}$=10). Linseed; three independent experimental replicates ($n_{ctrl}$ = 11 and $n_{3OHP}$=11). Dill; three independent experimental replicates ($n_{ctrl}$ = 14 and $n_{3OHP}$=13). Oregano; four independent experimental replicates ($n_{ctrl}$ = 40 and $n_{3OHP}$=39). Tomato; three independent experimental replicates ($n_{ctrl}$ = 11 and $n_{3OHP}$=15). A significant effect of treatment on the various species was tested by two-way ANOVA combining all the experimental replicates in a single model with treatment as a fixed effect and experiment as a random effect.

DOI: https://doi.org/10.7554/eLife.29353.007

*Figure 3 continued on next page*

*Figure 3 continued*

The following figure supplement is available for figure 3:

**Figure supplement 1.** Yeast response to 3OHP suggests a conserved target throughout eukaryotes.

DOI: https://doi.org/10.7554/eLife.29353.008

auxin over-production phenotypes as indicated by the *superroot* (*SUR*) 1 and 2 loci (*Mikkelsen et al., 2004*; *Boerjan et al., 1995*). However, the *SUR* genes are not evolutionarily conserved and 3OHPGSL does not create a superroot phenotype, showing that the genes are not the targets. A remaining conserved root regulator that does not alter meristem formation, but still alters root growth, is the TOR pathway (*Xiong et al., 2013*). Thus, we proceeded to test if mutants in the TOR pathway alter sensitivity to 3OHPGSL. Because TORC activity is sugar responsive, we investigated whether 3OHPGSL application may alter the response to sugar in genotypes with altered TORC activity. We first used the TOR kinase overexpression line GK548 (TORox) because it was the only one of several published TOR overexpression lines (*Deprost et al., 2007*) that behaved as a TOR overexpressor within our conditions (*Figure 5—figure supplement 1*). The GK548 TORox line exhibits accelerated TORC signaling and consequently grows longer roots on media containing sucrose (*Figure 5* and (*Deprost et al., 2007*)). In addition, GK548 TORox meristems are harder to arrest (*Figure 5B*). Applying 3OHPGSL to the GK548 TORox line showed that this genotype had an

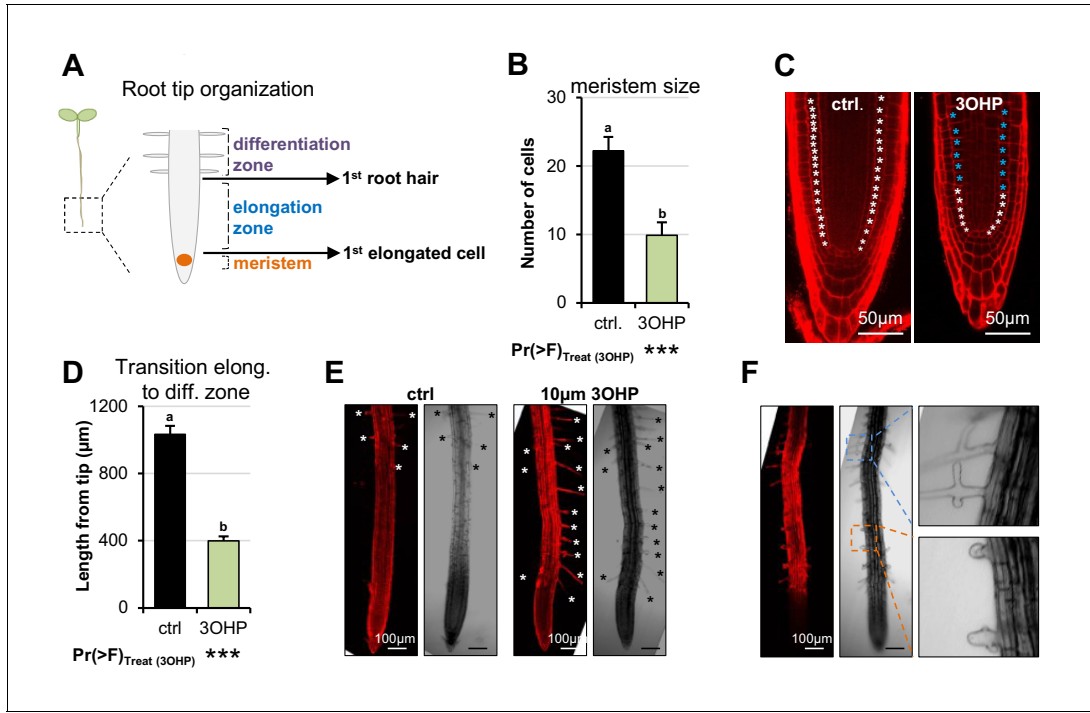

**Figure 4.** 3OHP reduces root zone sizes. (A) Diagrammatic organization of a root tip; the meristem zone from the QC to the first cell elongation; the elongation zone ends when first root hair appears (*Dolan and Davies, 2004*). (B) Meristem size of 4-d-old Arabidopsis seedlings grown on MS medium with sucrose ±10 µM 3OHP. Results are least squared means ± SE over three independent experimental replicates with each experiment having an average of three replicates per condition ($n_{ctrl}$ = 6; $n_{3OHP}$=9). Significance was tested via two-way ANOVA with treatment as a fixed effect and experiment as a random effect. (C) Confocal images of 4-d-old propidium iodide stained seedlings grown with and without 3OHP. Meristematic cells are marked with white asterisks, elongated cells with blue asterisks.( D) Appearance of first root hair; measured from the root tip on 4-d-old seedlings grown on MS medium with sucrose ±10 µM 3OHP. Results are least squared means ± SE over two independent experimental replicates with each experiment having an average of nine replicates per condition ($n_{ctrl}$ = 17; $n_{3OHP}$=20). Significance was tested via two-way ANOVA with treatment as a fixed effect and experiment as a random effect. (E) Confocal images of 4–d-old propidium iodide stained seedlings grown with and without 3OHP. Protruding root hairs are marked with white/black asterisks. (F) 3OHP induced root hair deformations, confocal images of 4–d-old propidium iodide stained seedlings grown with 3OHP.

DOI: https://doi.org/10.7554/eLife.29353.009

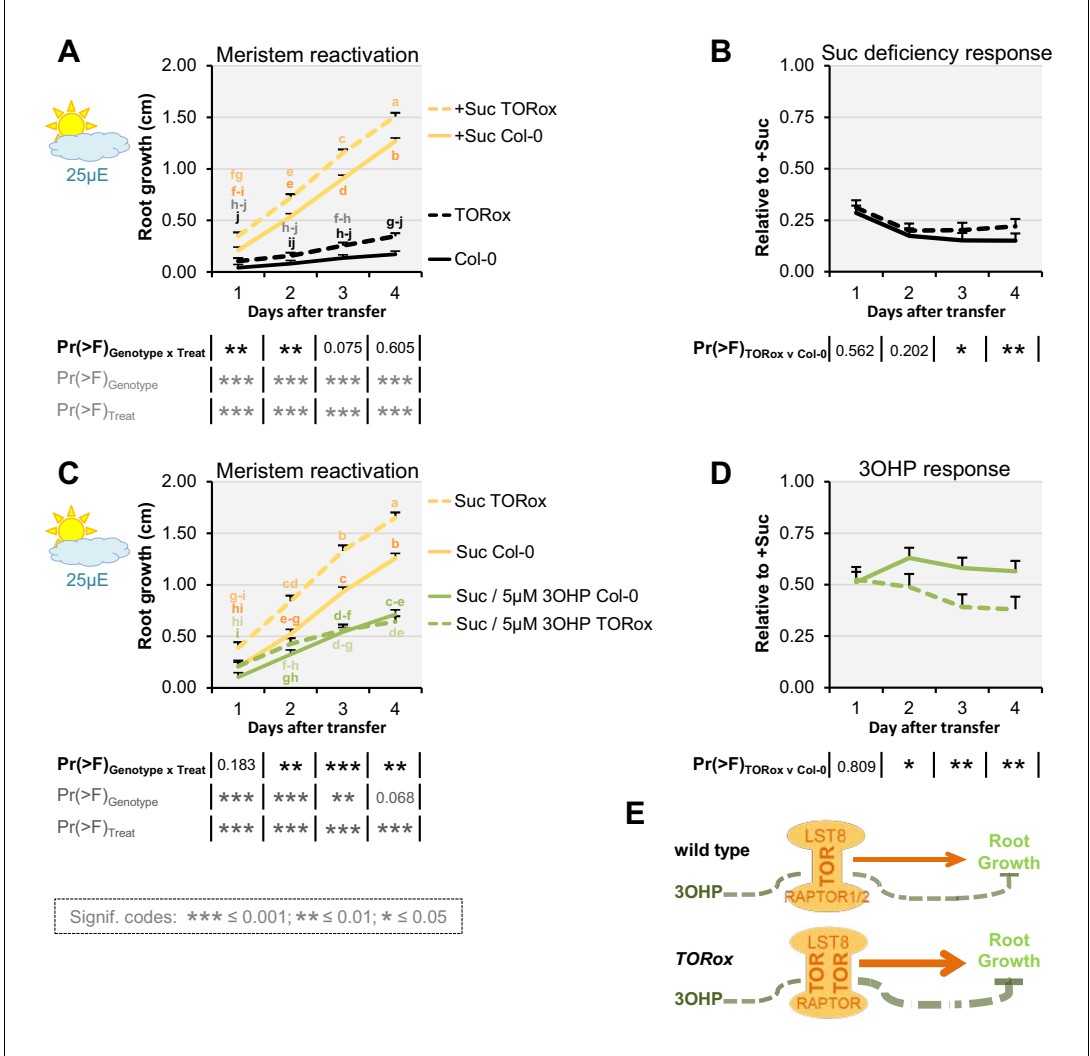

**Figure 5.** TOR over-activation amplifies 3OHP response. (**A**) Root growth for low light grown seedlings. The seedlings were grown on MS medium without sucrose for 3 days, then transferred to the indicated media (Suc; sucrose). Multi-factorial ANOVA was used to test the impact of Genotype (Col-0 v TORox), Treatment (Control v Sucrose) and their interaction on root length. All experiments were combined in the model and experiment treated as a random effect. The ANOVA results from each day are presented in the table. (**B**) The root lengths grown photo-constrained and without sucrose (from A) displayed at each time point as relative to the respective sucrose activated roots. Results least squared means ± SE over three independent experimental replicates with each experiment having an average of nine replicates per condition (n = 26–30). Multi-factorial ANOVA was used to test the impact of Genotype (Col-0 v TORox), Treatment (Sucrose v Sucrose/3OHP) and their interaction on root length. All experiments were combined in the model and experiment treated as a random effect. The ANOVA results from each day are presented in the table. (**C**) Root growth for low light grown seedlings. The seedlings were grown on MS medium without sucrose for 3 days, then transferred to the indicated media. (**D**) Photo-constrained root lengths in response to sucrose and 3OHP (from A) displayed at each time point as relative to the respective sucrose activated roots. Results are least squared means ± SE over two independent experimental replicates with each experiment having an average of six replicates per condition (n = 11–14). (**E**) Schematic model; over expression of the catalytic subunit TOR increases growth and the relative 3OHP response.

DOI: https://doi.org/10.7554/eLife.29353.010

The following figure supplements are available for figure 5:

**Figure supplement 1.** Published TORox lines that did not display the TORox phenotype under our conditions.
DOI: https://doi.org/10.7554/eLife.29353.011

**Figure supplement 2.** *RAPTOR1* haplo-insufficiency does not affect 3OHP response.
DOI: https://doi.org/10.7554/eLife.29353.012

**Figure supplement 3.** Loss of one of the two substrate-binding TORC-subunits affect 3OHP response.
DOI: https://doi.org/10.7554/eLife.29353.013

elevated 3OHPGSL-mediated inhibition of meristem reactivation in comparison to the WT (*Figure 5C–D*). This suggests that TORC activity influences the response to 3OHPGSL (*Figure 5E*).

We next investigated how genetically disrupting additional components of TORC affects 3OHPGSL responsiveness. In addition to the catalytic TOR kinase subunit, TORC consists of the substrate binding RAPTOR (*Henriques et al., 2014*; *Rexin et al., 2015*), and LST8 (13) (*Figure 5E*). In Arabidopsis- there is one copy of *TOR*, and two copies of both *RAPTOR* and *LST8* (*RAPTOR1/RAPTOR2* and *LST8-1/LST8-2*) (*Moreau et al., 2012*; *Deprost et al., 2005*).

RAPTOR1 and TOR null mutants are lethal as homozygotes (*Menand et al., 2002*; *Deprost et al., 2005*), and heterozygous *raptor1* mutants did not display a significant change in 3OHPGSL responsiveness (*Figure 5—figure supplement 2*). We therefore tested insertion mutants within the weaker homolog *RAPTOR2*, whose null mutant is viable, and in our conditions shows mildly reduced root length on sucrose-containing media (*Figure 5—figure supplement 3A and C*). We found that, for two independent insertion lines *raptor2-1* (*Deprost et al., 2005*; *Anderson et al., 2005*) and *raptor2-2* (*Deprost et al., 2005*) (*Figure 5—figure supplement 3E*), there was a statistically significant reduction in 3OHPGSL response (*Figure 5—figure supplement 3A–C*). This supports the hypothesis that 3OHPGSL-associated signaling proceeds through TORC and that *RAPTOR2* may play a stronger role in 3OHP perception than *RAPTOR1*.

## 3OHPGSL treatment inhibits sugar responses

A key function of TORC activity is to control meristem cell division and this can be measured by meristem reactivation assays (*Xiong et al., 2013*). Thus, to further test if TORC dependent responses are altered by 3OHPGSL, seedlings were germinated in sugar-free media and photosynthesis-constrained under low light conditions to induce root meristem arrest when the maternal glucose is depleted (three days after germination). The root meristems were reactivated by applying exogenous sucrose (*Figure 6A–C*). By treating arrested root meristems with sucrose alone or in combination with 3OHPGSL we found that 3OHPGSL could inhibit meristem reactivation of sugar-depleted and photosynthesis-constrained seedlings (*Figure 6D–E*). Further, this response was dependent upon the 3OHPGSL concentration utilized. A similar response was found when treating with a TOR inhibitor such as rapamycin (*Xiong et al., 2013*), providing additional support to the hypothesis that 3OHPGSL may reduce root growth by altering TORC activity.

## 3OHPGSL pharmacologically interacts with the TOR-inhibitor AZD-8055

To further examine the possibility that 3OHPGSL may be affecting the TOR pathway, we proceeded to compare the effect of 3OHPGSL to published chemical TOR inhibitors. The active site TOR inhibitors were originally developed for mammalian cells and inhibit root growth in various plant species (*Montané and Menand, 2013*). Similar to 3OHPGSL, the active-site TOR inhibitor AZD-8055 (AZD) induces a reversible concentration-dependent root meristem inhibition (*Montané and Menand, 2013*). By directly comparing 3OHPGSL treatment with known TOR chemical inhibitors in the same system, we can test for interactions between 3OHPGSL and the known TOR inhibitors. An interaction between 3OHPGSL application and a known TOR inhibitor, e.g. an antagonistic relationship, is an indication that the

same target is affected. To assess whether interactions between 3OHPGSL and TOR signaling occur, we grew seedlings vertically on media with combinations of 3OHPGSL and AZD and root-phenotyped the plants to compare the effect on root morphology. This identified a significant antagonistic interaction between AZD and 3OHPGSL (3OHP x AZD), both in terms of root length response (*Figure 7A*) and in initiation of the differentiation zone (*Figure 7B*). This antagonistic interaction is also supported by the appearance of first root hair (*Figure 7B*), as the premature initiation of the differentiation zone in the presence of 10 µM 3OHPGSL did not change further upon co-treatment (*Figure 7B*). Moreover, there was a vast overlap in the phenotypic response to both compounds (*Figure 7C–D*); notably the closer initiation of the root differentiation zone to the root tip (*Figure 7B–C*) and the decreased cell elongation (*Figure 7C*, right panel). Together, this suggests that the TOR inhibitor AZD and 3OHPGSL have a target in the same signaling pathway as no additive effect is observed. Supporting this is the observation that 3OHPGSL treatment is phenotypically similar to a range of TOR active site inhibitors, as well as an inhibitor of S6K1 (one of the direct targets of TOR) (*Figure 7—figure supplement 1*).

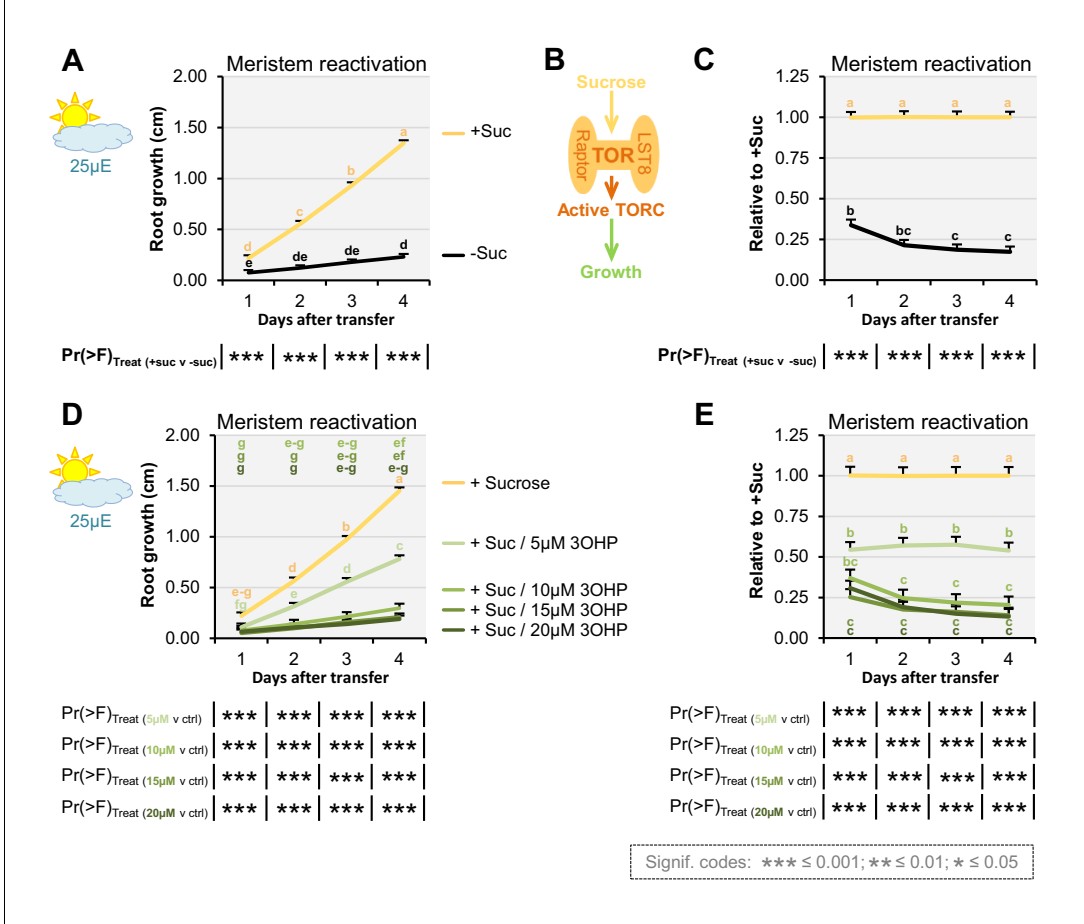

**Figure 6.** 3OHP dampens sugar-mediated meristem activation. (**A**) Root growth for low light grown Col-0 wildtype seedlings. The seedlings were grown on MS medium without sucrose for 3 days, then transferred to the indicated media. Multi-factorial ANOVA was used to test the impact of Treatment on root length. All experiments were combined in the model and experiment treated as a random effect. The ANOVA results from each day are presented in the table. (**B**) Schematic model; sucrose activates the TOR complex (TORC), leading to growth.( **C**) The root lengths (from A) displayed at each time point as relative to sucrose activated roots. Results are least squared means ± SE over five independent experimental replicates with each experiment having an average of eight replicates per condition (n_-Suc = 43; n_+Suc=40). (**D**) Root growth for low light grown seedlings. The seedlings were grown on MS medium without sucrose for 3 days, then transferred to the indicated media. Multi-factorial ANOVA was used to test the impact of Treatment on root length. All experiments were combined in the model and experiment treated as a random effect. The ANOVA results from each day are presented in the table. (**E**) The root lengths (from D) displayed at each time point as relative to sucrose activated roots (ctrl.). Results are least squared means ± SE over two independent experimental replicates with each experiment having an average of seven replicates per condition (n = 12–16).

DOI: https://doi.org/10.7554/eLife.29353.014

Interestingly, the short root hair phenotype induced by AZD showed a synergistic interaction between AZD and 3OHPGSL suggesting that they may target different components of the TORC pathway that interact (*Figure 7C*). Further, while there is strong phenotypic overlap between AZD and 3OHPGSL, there are also specific activities. AZD induced a rounding of the root tip (*Dolan and Davies, 2004*), but co-treatment with 3OHPGSL restored a wildtype-like tip phenotype (*Figure 7D*). The lack of root rounding and root hair inhibition suggest that AZD and 3OHPGSL both target the TOR pathway, but at different positions. Alternatively, the 3OHPGSL may be a more specific TOR inhibitor and the additional AZD phenotypes could be caused by the ATP-competitive inhibitor having alternative targets in plants. Together, these results suggest that 3OHPGSL directly or indirectly targets the same molecular pathway as known TOR inhibitors (*Jia et al., 2009*).

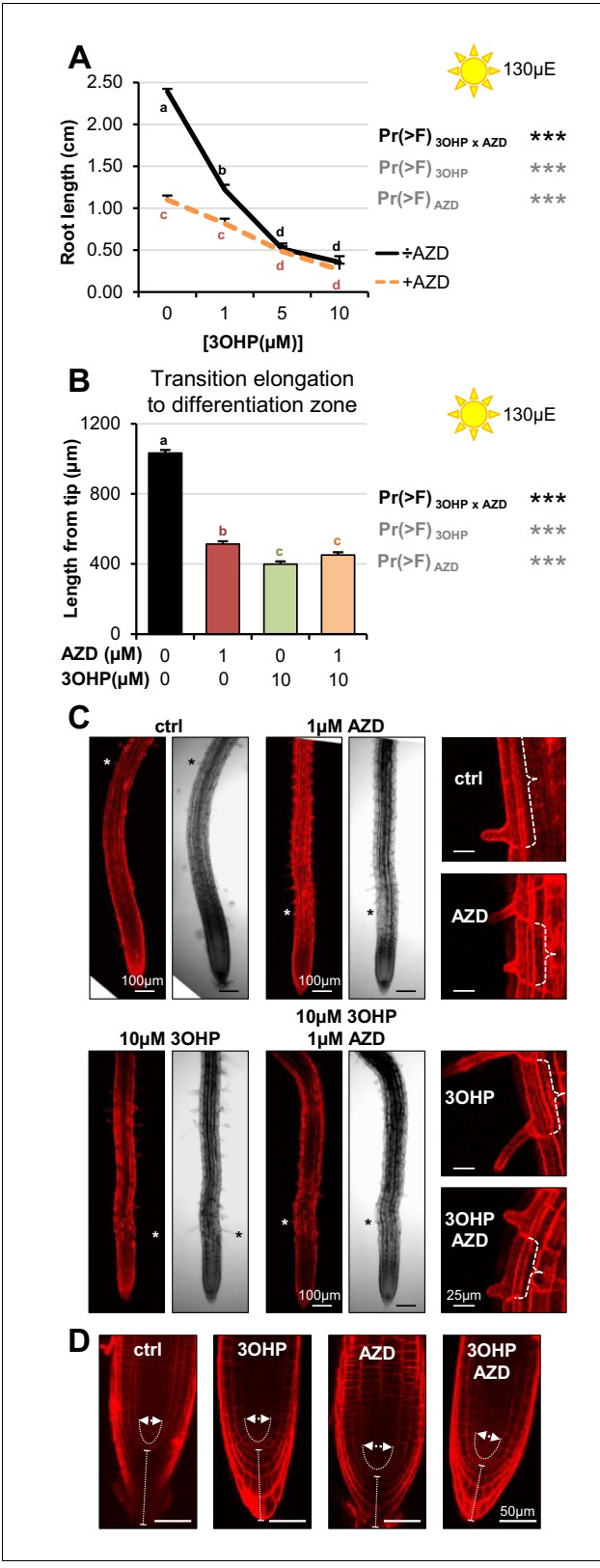

**Figure 7.** Pharmacological interaction of 3OHPGSL and the TOR inhibitor, AZD. (**A**) Root lengths of 7-d-old Col-0 wildtype seedlings grown on MS medium with sucrose ±combinations of AZD and different concentrations of 3OHP. Results are least squared means ± SE over three independent experimental replicates with each experiment having an average of nine replicates per condition (n = 18–58). Multi-factorial ANOVA was used to test

*Figure 7 continued on next page*

*Figure 7 continued*

the impact of the two treatments and their interaction on root length. All experiments were combined in the model and experiment treated as a random effect. The ANOVA results from each day are presented in the table. (B) Appearance of first root hair; measured from the root tip on 4–d-old seedlings grown on the indicated MS medium with sucrose. Results are least squared means ± SE over two independent experimental replicates with each experiment having an average of nine replicates per condition (n = 17–20). Multi-factorial ANOVA was used to test the impact of the two treatments and their interaction on root length. All experiments were combined in the model and experiment treated as a random effect. The ANOVA results from each day are presented in the table. (C) Confocal images of 4-d-old propidium iodide stained seedlings. The first protruding root hairs are marked with white/black asterisks on the left panel. Right panel shows zooms of first root hair, cell size is indicated. (D) Confocal images of 4–d-old propidium iodide stained seedlings.

DOI: https://doi.org/10.7554/eLife.29353.015

The following figure supplement is available for figure 7:

**Figure supplement 1.** Pharmacological interactiosn between 3OHPGSL and diverse TOR inhibitors.

DOI: https://doi.org/10.7554/eLife.29353.016

## Blocking parts of the autophagy machinery affects 3OHPGSL associated signaling

Activation or repression of the TOR pathway leads to regulatory shifts in numerous downstream pathways (*Figure 8E*) (*Sheen, 2014*; *Sablowski et al., 2014*). For example, active TOR negatively regulates autophagy across eukaryotic species including Arabidopsis (*Liu and Bassham, 2010*; *Shibutani and Yoshimori, 2014*). To test if pathways downstream of TORC are affected by, or involved in, 3OHPGSL signaling, we analyzed mutants of two key autophagic (ATG) components, *atg2-1* (18) and *atg5-1* (58). ATG2 is part of the ATG9 cycling system that is essential for autophagosome formation (*Feng et al., 2014*; *Velikkakath et al., 2012*; *Ryabovol and Minibayeva, 2016*). ATG9-containing vesicles are a suggested membrane source for the autophagosome, and vesicles containing ATG9 are cycled to-and-from the phagophore via the ATG9 cycling system (*Feng et al., 2014*; *Ryabovol and Minibayeva, 2016*). ATG5, is part of the dual ubiquitin-like conjugation systems responsible for ATG8 lipidation (*Feng et al., 2014*; *Ryabovol and Minibayeva, 2016*; *Fujioka et al., 2008*). There are nine *ATG8* paralogues in Arabidopsis (*Ryabovol and Minibayeva, 2016*) and together with the single copy of *ATG5*, they are essential for autphagosome initiation, expansion, closure, and vacuolar fusion (*Feng et al., 2014*; *Ryabovol and Minibayeva, 2016*). After the first conjugation system has conjugated ATG8 to an E2-like enzyme, the E3 ligase-like activity of the second ATG5-containing system enables ATG8 lipidation at the autophagic membrane (*Feng et al., 2014*; *Walczak and Martens, 2013*; *Kaufmann et al., 2014*). We found that *atg5-1* enhanced 3OHPGSL responsiveness (*Figure 8A–B*) while *atg2-1* had a wild type response (*Figure 8C–D*). One possible explanation for this difference between the two mutants is that, apart from macro-autophagy, plants also have micro-autophagy (*Ryabovol and Minibayeva, 2016*), a process that, in animal systems, has been shown to be negatively regulated by TOR (*Li et al., 2012*). Micro-autophagy does not involve de novo assembly of autophagosomes, and ATG5 has been shown to be involved in several forms of micro-autophagy whereas the role of ATG2 is more elusive and may not be required (*Li et al., 2012*). Thus, the elevated 3OHPGSL response in the *atg5-1* mutant supports the hypothesis that 3OHPGSL signaling proceeds through the TOR pathway, but also suggests that this response requires parts of the autophagic machinery as it was not observed for *atg2-1*.

## Discussion

In this study we describe a novel signaling capacity associated with 3OHPGSL, a defense metabolite present in Arabidopsis, and provide evidence that the linked signal proceeds via the TOR pathway. Application of exogenous 3OHPGSL caused reversible root meristem inhibition by morphological reprogramming of the root zones, i.e. dramatically reduced the root meristem size and limited root cell elongation (*Figure 4*). This response occurred at levels within the endogenous range and there was no evidence of cell death in any treated root, suggesting that this is not a toxicity response (*Figure 1*). Additionally, these morphological responses were specific to 3OHPGSL and not caused by

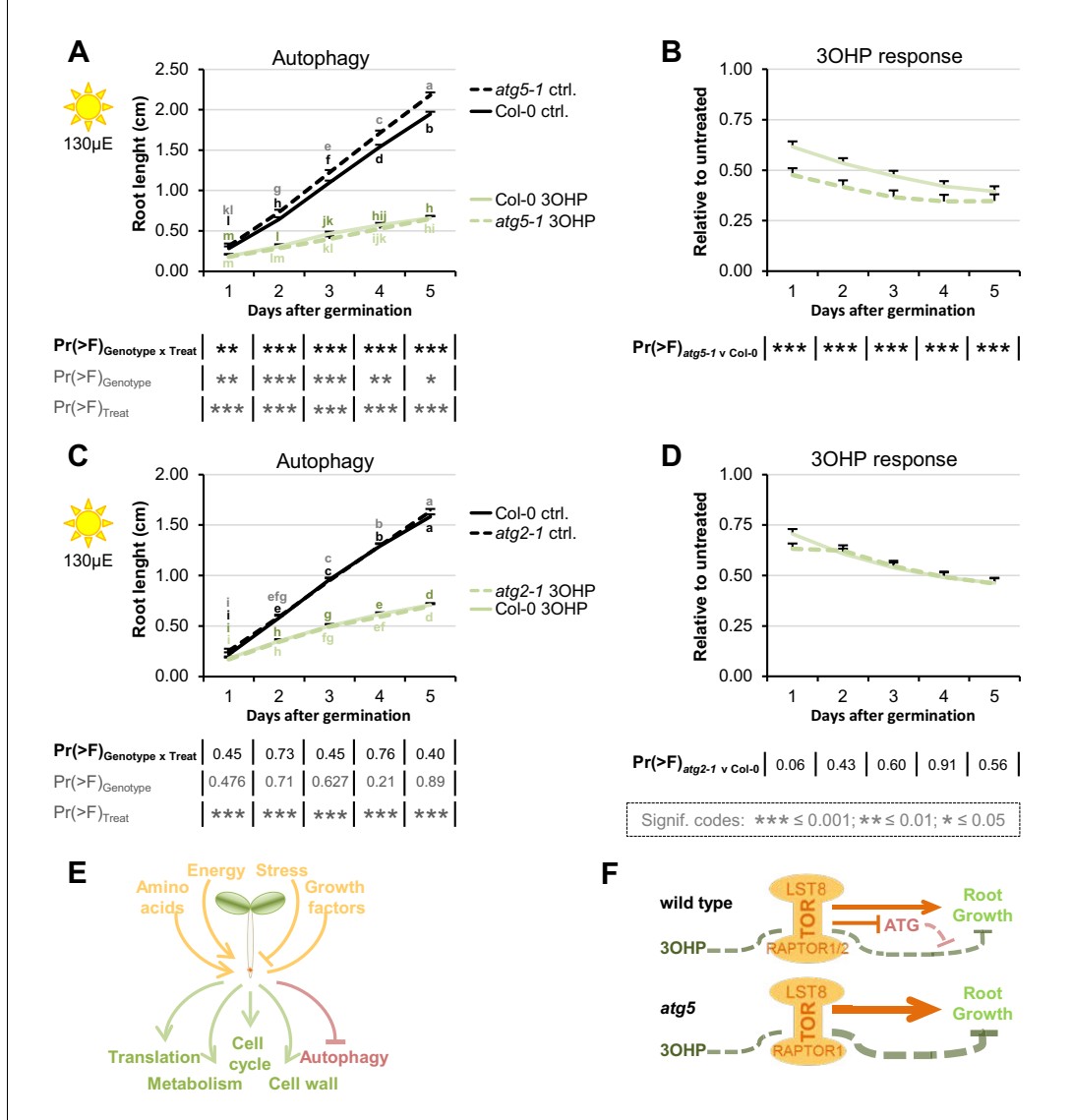

**Figure 8.** Blocking autophagosome elongation amplifies the 3OHP response. (**A**) Root growth for *atg5-1* and wildtype Col-0 seedlings grown on MS medium supplemented with or without 5 µM 3OHP. Multi-factorial ANOVA was used to test the impact of Genotype (Col-0 v *atg5-1*), Treatment (Control v 3OHP) and their interaction on root length. All experiments were combined in the model and experiment treated as a random effect. The ANOVA results from each day are presented in the table. (**B**) Root lengths in response to 3OHP (from A) displayed at each time point as relative to untreated. Results are least squared means ± SE over two independent experimental replicates with each experiment having an average of 21 replicates per condition (n = 31–52). (**C**) Root growth for *atg2-1* and wildtype Col-0 seedlings grown on MS medium supplemented with or without 5 µM 3OHP. Multi-factorial ANOVA was used to test the impact of Genotype (Col-0 v *atg5-1*), Treatment (Control v 3OHP) and their interaction on root length. All experiments were combined in the model and experiment treated as a random effect. The ANOVA results from each day are presented in the table. (**D**) Root lengths in response to 3OHP treatment (from C) displayed at each time point as relative to untreated. Results are least squared means ± SE over two independent experimental replicates with each experiment having an average of 26 replicates per condition (n = 36–66). (**E**) The TOR complex (TORC), is affected by several upstream input, leading to activation or repression of several downstream pathways. (**F**) Schematic model; sucrose activates TORC, leading to root growth. 3OHP represses root growth through interaction with TORC. Autophagy pathways via ATG5 negatively affect 3OHP response.

DOI: https://doi.org/10.7554/eLife.29353.017

any structurally or biosynthetically related GSL, suggesting that these responses were not because of generic properties shared by GSLs (*Figure 2*). Exposing a wide phylogenetic array of plants, including lineages that have never produced GSLs, to 3OHPGSL showed that application of this compound can inhibit growth broadly across the plant kingdom as well as in yeast (*Figure 3*,

*Figure 3—figure supplement 1*). This suggests conservation of the downstream signaling pathway across these diverse plant lineages. Equally, if the signaling compound is not 3OHPGSL itself, but a derivative, then the required biosynthetic processes must be conserved. This conservation largely rules out the specific GSL activation pathway controlled by Brassicales specific thioglucosidases, myrosinases (*Barth and Jander, 2006*; *Nakano et al., 2017*; *Bones and Rossiter, 2006*). The phylogenetic conservation of the 3OHPGSL response led us to search for a target pathway controlling growth and development that would be evolutionary well conserved between the tested species.

By comparing the root phenotype identified with 3OHPGSL application to the published literature, we hypothesized that 3OHPGSL treatment may affect TORC, a key primary metabolic sensor that controls growth and development, and is conserved back to the last common eukaryotic ancestor (*Henriques et al., 2014*). Active site TOR inhibitors inhibit root growth in numerous plant species similar to 3OHPGSL application (*Montané and Menand, 2013*), supporting the hypothesis that 3OHPGSL may function via TORC. A model with 3OHPGSL affecting TORC would explain how 3OHPGSL can alter root development across the plant kingdom (*Figure 3*). Mechanistic support for this hypothesis came from a number of avenues. First, 3OHPGSL can block the TOR-mediated sugar activation of arrested meristems (*Figure 6*). Second, the TORox mutant intensifies 3OHPGSL linked signaling (*Figure 5*), and correspondingly loss-of-function mutants of the substrate binding TORC component *raptor2* diminish the 3OHPGSL effect (*Figure 5—figure supplement 3*). Additionally, there are clear phenotypic overlaps between the root phenotypes induced by known TOR inhibitors and 3OHPGSL, e.g. root inhibition, inhibition of cell elongation, and notably the dramatic reduction of the meristem sizes (*Figure 7*). Critically, 3OHPGSL and known small-molecule inhibitors of TOR were mutually antagonistic for a number of phenotypes. In pharmacology, the outcomes of a drug combination can either be antagonistic, additive or synergistic, depending on whether the effect is less than, equal to, or greater than the sum of the effects of the two drugs (*Jia et al., 2009*). Antagonistic interactions, as observed with 3OHPGSL and AZD, can occur if two drugs exhibit mutual interference against the same target site, or if their targets converge on the same regulatory hub (*Jia et al., 2009*). Together, these lines of evidence suggest that 3OHPGSL or a derived metabolite targets the TOR pathway to alter root meristem development within Arabidopsis and potentially other plant species.

Extending the analysis to pathways downstream of TORC, showed that loss of ATG5, a vital component of the autophagic machinery (*Liu and Bassham, 2010*; *Shibutani and Yoshimori, 2014*), intensifies the 3OHPGSL response (*Figure 8A*-DB). This supports the hypothesis that 3OHPGSL signaling proceeds through the TOR complex, but also suggest that this signal requires parts of the autophagic machinery. Loss of another autophagic component, ATG2, did not influence the 3OHP response (*Figure 8C–D*). Together, this raises the possibility that 3OHPGSL influenced responses involve predominantly a micro-autophagy pathway, which is ATG5- but may not be ATG2-dependent, rather than the macro-autophagy pathway that depends upon both genes (*Ryabovol and Minibayeva, 2016*; *Li et al., 2012*). Micro-autophagy removes captured cytoplasmic components directly at the site of the vacuole via tonoplast invagination. The cargo to be degraded ranges from non-selective fractions of the cytoplasm to entire organelles, dependent on the type of micro-autophagy. The two ubiquitin-like conjugation systems, and thereby ATG5, have been shown to be involved in several forms of micro-autophagy, such as starvation-induced, non-selective, and glucose-induced selective autophagy (*Li et al., 2012*). Interestingly, micro-autophagy involves vacuolar movement of cargo, and the vacuole is considered the main storage site for glucosinolates (*Mikkelsen et al., 2004*). Thus, ATG5 may be responsible for enabling the movement of exogenously applied 3OHPGSL out of the cytoplasm where it or a derivative metabolite could interact with the TORC pathway and into the vacuole. This would decrease the concentration of the 3OHPGSL associated signal and could explain why the *atg5-1* mutant is more sensitive to 3OHPGSL application. Further work is required to test if ATG5 is functioning to attenuate the 3OHPGSL associated signal.

A conundrum for defense signaling compounds to affect growth is the evolutionary age discrepancy; defense metabolites are typically evolutionarily very young, as they are often species or taxa specific, while growth regulatory pathways are highly conserved across broad sets of plant taxa. This raises the question of which mechanism(s) may allow this connection between young metabolites and old regulatory pathways. This suggests that plants may sense young metabolites using evolutionarily old signaling pathways. Similar evidence is coming from other secondary metabolite

systems suggesting that this may be a general phenomenon. For example, an indolic GSL activation product can interact with the conserved *TIR1* auxin receptor to alter auxin sensitivity within Arabidopsis (*Katz et al., 2015*). Similarly, an unknown phenolic metabolite appears to affect regulation of growth and development by influencing the Mediator complex that is conserved across all eukaryotes (*Bonawitz et al., 2012*; *Bonawitz et al., 2014*; *Kim et al., 2014*); and the plant polyphenol resveratrol directly inhibits the mammalian TOR to induce autophagy (*Park et al., 2016*). Thus, young plant metabolites can influence evolutionarily conserved pathways. Interestingly, this strongly resembles the action of virulence-associated metabolites within plant pathogens. Pseudomonad bacteria produce the evolutionarily young coronatine that alters the plant defense response by interacting with the conserved JA-Ile receptor COI1 (*Xie et al., 1998*). In plant/pathogen interactions, this ability of pathogen-derived metabolites to alter plant defense signaling is evolutionarily beneficial because it boosts the pathogen's virulence *in planta*. It is less clear if this selective pressure model also applies to plant defense compounds that interact with endogenous signaling pathways. Following the plant/pathogen derived model, it is tempting to assume that such plant defense metabolites have been co-selected on their ability to affect the biotic attacker and simultaneously provide information to the plant. However, an alternative hypothesis that is that these examples may simply be serendipitous cases, where the defense metabolites happened to interact with a pathway and are potentially of no evolutionary benefit. In this model, the plant might still be adapting to the evolution of this new regulatory linkage. In the particular case of 3OHPGSL, the AOP3 enzyme that makes this compound evolved prior to the split between *A. thaliana* and *A. lyrata* suggesting that these species have had at least several million years/generations of potential to adapt. However, the two hypotheses need to be empirically tested. Central to testing between the two hypotheses is to assess if the observed signaling effects have any fitness benefit for the plant suggesting that even if the connections arose by serendipity that they have been maintained by a selective benefit. This will require field testing the fitness of plants that contrast for the presence of these connections. An alternate way to test between these hypotheses would be to conduct a broad survey of plant metabolites to test how many can affect signaling within the plant. If a large fraction of metabolites have potential signaling function, it is unlikely that all of these are simply serendipitous cases that have not had sufficient time to be removed by natural selection. However; earlier studies have provided evidence that both allyl GSL and the GSL breakdown product indole-3-carbinol affect plant signaling and growth (*Katz et al., 2015*; *Francisco et al., 2016a*; *Francisco et al., 2016b*). In addition, *R*- and *S*-2-hydroxybut-3-enyl GSL promoted root growth (*Figure 2—figure supplement 1*), suggesting that dual effects of defense metabolites such as 3OHPGSL are possibly more general.

Within this report, we provided evidence that 3OHPGSL, or derived compounds, appears to function as a natural endogenous TORC inhibitor that can work across plant lineages. This creates a link whereby the plant's endogenous defense metabolism can simultaneously coordinate with growth. Such a built-in signaling capacity would allow coordination between development and defense, as the plant could use the defense compound itself as a measure of the local progress of any defense response and readjust development and defense to optimize against the preeminent threat. Future work is required to identify the specific molecular interaction that allows this communication to occur, this will help to illuminate how and why plants measure their own defense metabolism to coordinate available resources more broadly with growth. Future work might also ascertain whether there is a broader class of plant produced TOR inhibitors. If this is true, they might be highly useful in understanding TOR function across kingdoms of life and possibly to reveal significant aspects of this universally conserved pathway that may have gone unnoticed in other eukaryotic models.

## Materials and methods

### Plant materials

The genetic background for the Arabidopsis (*Arabidopsis thaliana*) mutants and transgenic lines described in this study is the Col-0 accession. The following lines were described previously: *myb28-1 myb29-1* (*SonderbySønderby et al., 2007*), *atg2-1* (*Inoue et al., 2006*), *atg5-1* (58*Thompson et al., 2005*), *raptor1-1* (*Deprost et al., 2005*; *Anderson et al., 2005*), *raptor1-2* (*Deprost et al., 2005*), *raptor2-2* (*Deprost et al., 2005*), *raptor2-1* (*Deprost et al., 2005*; *Anderson et al., 2005*), and the *TORox* lines *G548, G166, S784,* and *S7817* (*Deprost et al., 2007*).

All genotypes were obtained and validated both genetically and phenotypically as homozygous for the correct allele.

## Plant growth media and in vitro root growth assays

Seeds were vapour sterilized for 2–3 hr, by exposure to a solution of 100 mL household bleach (Klorin Original, Colgate-Palmolive A/S) mixed with 5 mL hydrochloric acid (12M), and ventilated for 30 min to one hour. After plating, on ½ strength Murashige and Skoog (MS) medium (2.2 g/l MS +vitamins) (Duchefa) with 1% (w/v) sucrose (Nordic Sugar), and 0.8% (w/v) micro agar (Duchefa), pH adjusted to 5.8), the seeds were stratified for two days in the dark at 4°C. For root length assays at normal light (115-130 µE) Arabidopsis seedlings were grown vertically at 22 °C day 20°C night under a 16 hr photoperiod and 80% humidity (long day). Concentrated 3OHPGSL in water was added to the agar post-sterilization to create media with the described concentration for each assay. The same method and water was used to create the media for the testing of the other specific GSLs. For meristem reactivation assays plants were grown as described in (*Xiong et al., 2013*), except that in our conditions we needed to go to 25µE to obtain meristem inhibition. Daily root lengths were manually marked (from day 3) with a permanent marker pen on the backside of the plate. After photography of 7-d-old seedlings the root growth was quantified using the ImageJ software (*Schneider et al., 2012*). The least square means (lsmeans) for the genotypes in response to different treatments were calculated across experiments (in R, see statistics), and plotted in excel.

## In vitro root growth assays for the species (seed plating and growth conditions)

To test 3OHPGSL perception in other plant orders, seeds were obtained as listed in *Supplementary File 1*. Except for *Solanum lycopesicum,* here *San Marzano* tomatoes were bought in a local supermarket and the seeds were harvested, fermented and dried. All seeds were vapour sterilized for three hours (as above). Before plating, and *Lotus japonicus MG20* (Lotus) were emerged in water and kept at 4°C for 1–2 weeks. Seeds were plated on vertical ½MS plates as specified in *Supplementary File 1*, stratified for four days in the dark at 4°C before being transferred to a long day growth chamber). Root growth was measured approximately every 24 hr (as described above).

## Yeast strain, media, and growth conditions

The yeast strain, NMY51 with pOST1-NubI and pDHB1-LargeT ((*Stagljar et al., 1998*; *Möckli et al., 2007*); DUALsystem Biotech), was grown in liquid YPD media (2% w/v bactopeptone (Duchefa Biochemie), 1% w/v yeast extract (Becton, Dickinson and Company), 2% w/v glucose) with or without added GSLs, at 30°C and 150 rpm shaking.

## Yeast growth assay

On day one; a 5 ml overnight culture was started from cryostock. Day two; four new 4 ml cultures were inoculated with 1 ml overnight culture, and grown overnight. On day three; an OD600 0.4 and a 0.04 dilution was prepared from each of the four cultures. 500 µl of each of the four cultures, at both dilutions, were transferred to a 96-well culture plate containing 500 µl YPD liquid media with 3OHPGSL or Allyl GSL, to final OD600 0.2 and GSL concentrations of 50, 10, 5, 1 and 0 µM. The yeast growth was measured at 0, 4, 6, 8, 24 and 48 hr. For each growth measurement 100 µl culture was transferred to a 96-well Elisa-plate together with three wells of YPD liquid media for standardization. Growth was measured with a SpectraMAX 190 (Molecular Devices) and SoftMax Pro 6.2.2 software. Growth rates and statistical analysis was calculated using the R software. The linear growth range was determined, and a linear regression using the lm() function in R was carried out to determine OD600 increase per hour (slope) and the yeast doubling time was calculated.

## Glucosinolate analysis

Glucosinolates were extracted from whole plant tissue of adult plants (for 3OHPGSL extraction), or from or 10-d-old seedlings (3OHPGSL uptake) (*Kliebenstein et al., 2001a*; *Kliebenstein et al., 2001b*; *Kliebenstein et al., 2001c*), and desulfo-glucosinolates were analysed by LC-MS/TQ as desulfo-GSLs as described in (*Crocoll et al., 2016*).

## Statistics

The R software with the R studio interface was used for statistical analysis (*R Core Team, 2017*; *RStudio Team, 2015*). Significance was tested using the Anova function (aov), lsmeans were obtained using the 'lsmeans' package (version 2.17) (*Lenth, 2016*). The letter groupings (Tukey's HSD Test) were obtained using the 'agricolae' package (version 1.2–3) (*de Mendiburu, 2010*).

## Confocal microscopy

To examine the root tip zones, we used confocal laser-scanning microscopy of 4-d-old seedlings grown vertically with or without treatment (with 3OHPGSL and/or various inhibitors). Samples were mounted on microscopy slides in propidium iodide solution (40 μM, Sigma) and incubated for 15 min. Confocal laser scanning microscopy was carried out on a Leica SP5-X confocal microscope equipped with a HC PL FLUOTAR 10 DRY (0.3 numerical aperture, 10X magnification) or a HCX lambda blue PL APO 320 objective (0.7 numerical aperture, 20X magnification) for close-up pictures of the meristem. To visualize the cell walls of individual cells the propidium iodide stain was excited at 514 nm and emission was collected at 600 nm to 680 nm. To determine the size of the meristems the confocal pictures we manually inspected and the meristematic cells marked and counted (the meristem region is defined as in (*Dolan and Davies, 2004*; *Perilli and Sabatini, 2010*)). To measure the distance from the root tip to the point of first root hair emergence we used ImageJ (*Schneider et al., 2012*).

## Chemicals

The AZD8055 (*Chresta et al., 2010*), Torin2 (*Liu et al., 2011*), KU-63794 (*García-Martínez et al., 2009*), and WYE-132 (*Yu et al., 2010*) were purchased from Selleckchem. PF-4708671 (*Pearce et al., 2010*) and allyl/sinigrin were purchased from Sigma-Aldrich. 4MSB and 3MSP GSLs were purchased at C2 Bioengineering. But-3-enyl GSL was purified from *Brassica rapa* seeds while 3OHPGSL was purified from the aerial parts of 4–5 weeks old greenhouse-grown plants of the Arabidopsis accession Landsberg *erecta* (*Kliebenstein et al., 2001c*; *Crocoll et al., 2016*). The concentration of 3OHP and but-3-enyl GSL was determined by LC-MS/TQ as desulfo-GSLs. All inhibitors were dissolved in DMSO and stored as 10 mM stocks at –20°C. For allyl, 3MSP, and 4MSB ~ 100 mM GSL stocks were made with H2O and the concentration of GSLs within these stocks was determined by LC-MS/TQ (see above).

## Acknowledgements

We thank the excellent technical assistance of the PLEN Greenhouse staff, and the DynaMo student helpers. We thank Dr. Svend Roesen Madsen for providing Camelina, and rape seeds, and Dr. Camilla Knudsen Baden for giving us lotus seeds.

## Additional information

### Competing interests

Daniel J Kliebenstein: Reviewing editor, *eLife*. The other authors declare that no competing interests exist.

### Funding

| Funder | Grant reference number | Author |
|---|---|---|
| Danmarks Grundforsknings-fond | DNRF99 | Frederikke Gro Malinovsky<br>Marie-Louise F Thomsen<br>Sebastian J Nintemann<br>Baptiste Bourgine<br>Meike Burow<br>Daniel J Kliebenstein |
| National Science Foundation | IOS 13391205 | Daniel J Kliebenstein |
| National Science Foundation | MCB 1330337 | Daniel J Kliebenstein |

| U.S. Department of Agriculture | CA-D-PLS-7033-H | Daniel J Kliebenstein |

The funders had no role in study design, data collection and interpretation, or the decision to submit the work for publication.

## Author contributions
Frederikke Gro Malinovsky, Conceptualization, Data curation, Formal analysis, Validation, Investigation, Visualization, Methodology, Writing—original draft, Writing—review and editing; Marie-Louise F Thomsen, Formal analysis, Validation, Investigation, Writing—review and editing; Sebastian J Nintemann, Lea Møller Jagd, Investigation, Visualization, Writing—review and editing; Baptiste Bourgine, Investigation, Writing—review and editing; Meike Burow, Conceptualization, Resources, Formal analysis, Supervision, Funding acquisition, Investigation, Visualization, Methodology, Writing—original draft, Project administration, Writing—review and editing; Daniel J Kliebenstein, Conceptualization, Data curation, Formal analysis, Supervision, Funding acquisition, Visualization, Methodology, Writing—original draft, Project administration, Writing—review and editing

## Author ORCIDs
Frederikke Gro Malinovsky http://orcid.org/0000-0002-9833-7968
Meike Burow http://orcid.org/0000-0002-2350-985X
Daniel J Kliebenstein http://orcid.org/0000-0001-5759-3175

## Decision letter and Author response
Decision letter https://doi.org/10.7554/eLife.29353.021
Author response https://doi.org/10.7554/eLife.29353.022

## Additional files

### Supplementary files
• Supplementary file 1. Plant seeds used for in vitro root growth assays for the various plant species. The source, common name, order, family, genus, species, and subspecies seeds are listed for the used plant species. As well as the plate size (cm × cm), and plating distance used for the individual response assays.
DOI: https://doi.org/10.7554/eLife.29353.018

• Transparent reporting form
DOI: https://doi.org/10.7554/eLife.29353.019

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
