## [Decision Letter]

Thank you for submitting your article "An evolutionary young defense metabolite influences the root growth of plants via the ancient TOR signaling pathway" for consideration by *eLife*. Your article has been reviewed by two peer reviewers, and the evaluation has been overseen by a Reviewing Editor and Ian Baldwin as the Senior Editor. The following individuals involved in review of your submission have agreed to reveal their identity: John Pickett (Reviewer #1); Reuben Peters (Reviewer #2).

The reviewers have discussed their comments with one another and with the Reviewing Editor. The individual reviews are following below. As an outcome of the consultation session, the reviewers and the Reviewing Editor concluded that the additional work requested by reviewer #1, and in particular items 1 and 2, should be included in a revised submission. In addition please consider all other comments to improve the paper.

*Reviewer #1:*

This is an excellent piece of work by a highly competent team and in essence provides evidence of a newly observed and unexpected interaction with the pathway TOR (Target Of Rapamycin) which is conserved across the living kingdoms of eukaryotes, and in this work studied for plants and fungi. The work included is extensive and represents an extremely conscientious effort logically to develop arguments for the importance of the discovery. However, although the work stands without question, as offered in this paper, some of the conclusions particularly those relating to the evolutionarily recent signalling role of the particular compound, 3-hydroxypropylglucosinolate (3OHPGSL), observed making this novel interaction, there are further considerations requiring more experimental work that I believe are necessary before such conclusions can be made with confidence. First of all it will be noted that I have run together the name of this compound, which is essential in describing the glucosinolates central to this publication. The particular glucosinolate is not a salt of the 3-hydroxypropyl ion nor is it an ester like molecule as in 3-hydroxypropyl acetate but is a glucosinolate substituted by 3-hydroxypropyl.

Turning now to the evolutionarily related arguments and conclusions, these may well be correct but, in my view, do not stand without the following considerations and related new experimental work. Because the activity of 3OHPGSL is found beyond the plant order Brassicales then the authors are correct in eliminating the potential role of specific catabolitic enzymes acting on the glucosinolates in generating active species. 1) However, they have not proved satisfactorily that it is the intact 3OHPGSL molecule that has the observed activity. 2) If the study was extended to, for example 2-hydroxy-3-butenylglucosinolate then they would have a glucosinolate example more closely related structurally to 3OHPGSL. Also this would provide an example of a compound similar to 3OHPGSL in that it is potentially capable of undergoing related chemical changes which could take place beyond the Brassicales. 3) A further study, though less essential than 1 and 2, would be to investigate cyanogenic glycocides because these are considered to be the evolutionary ancestors of the glucosinolates. This would set in a more clear perspective the assertions regarding the recent evolution of the glucosinolates

*Reviewer #2:*

This manuscript describes the intriguing finding that a specific secondary metabolite, 3-hydroxypropyl glucosinolate (3OHP GSL), can inhibit the universal eukaryotic TOR signaling complex, which the producing plant *Arabidopsis thaliana* appears to have adapted to use as a regulatory measure for balancing growth and development with the available energy and nutrients. The evidence for this use is strong, including identification of the target as TOR itself, and the authors have further investigated the downstream signaling pathway and effect, which appear to act on autophagy. However, it is less clear that this is a generalizable regulatory mechanism, which the authors suggest might be the case. Although softened to some extent with appropriate caveats within the Discussion, this is perhaps oversold in the Abstract. One issue that is perhaps not fully discussed is the uncertain means by which *A. thaliana* has adapted to this effect. The generality of the effect of 3OHP GSL on TOR signaling in other, non-producing plants, as well as yeast, demonstrate that TOR itself is conserved. Moreover, given the similar phenotypic effect of 3OHP GSL on other plant species, it is not entirely clear how much adaptation has actually occurred versus selection for (or at least a lack of selection against) the (apparently) serendipitous TOR inhibition exhibited by this natural product. It would be helpful if the authors include some discussion of this point, as well as be a bit more realistic about the use of evolutionary young defense metabolites as regulators, which seems almost certain to require presumably serendipitous activity against already extant signaling pathways (although another possibility is that this activity is related to their defensive role, which would require proof of greater activity against the 'target' versus 'native producer' organism). It seems likely that the controls included application of the carrier solution, but this is not immediately evident in the Materials and methods description, and needs to be clarified.

---

## [Author Response]

The reviewers have discussed their comments with one another and with the Reviewing Editor. The individual reviews are following below. As an outcome of the consultation session, the reviewers and the Reviewing Editor concluded that the additional work requested by reviewer #1, and in particular items 1 and 2, should be included in a revised submission. In addition please consider all other comments to improve the paper.

Items 1 and 2 were focused on providing additional evidence for what may or may not be the bioactive compound. As suggested by the reviewers, we have conducted separate experiments with the R and S enantiomers of 2hydroxy-but-3-enylglucosinolate. These showed that neither compound has a similar repressive effect on root elongation and if anything have a stimulatory effect on root elongation. As such, we can conclude that it is not the simple presence of a hydroxyl moiety that leads to the activity.

We believe that these new experiments help to refine the potential bioactive component but agree that we have not absolutely proven that it is the 3-hydroxybutylglucosinolate that is mediating the final signal transduction. We were very careful not to make this claim. This is especially the case, because the only way to provide this evidence is to identify the RNA or protein with which the metabolite interacts to stimulate the signal transduction. For example, it was not until the COI1/JAZ/JA-ILE structure was identified that it was absolutely clear that JA-Ile and not JA or Me-JA is the signal. Similarly, it was cloning of the TIR1/AFB genes that showed why most short acetic acids produce auxin like activities because the TIR1/AFB receptors appear to have different binding affinity. We agree that it is important to find the receptor molecule but feel that it took decades to find the receptor for JA, SA, Auxin, etc. and that this is beyond the scope of a single paper. To recognize this, we had taken great care in commenting that 3-hydroxybutylglucosinolate can stimulate the response but may not be the bioactive metabolite. We hope that this is acceptable and that the reviewers and editors agree that showing that a glucosinolate leads to altered development via an unknown interaction with the TOR pathway is sufficiently novel to merit inclusion in *eLife*.

Reviewer #1:

This is an excellent piece of work by a highly competent team and in essence provides evidence of a newly observed and unexpected interaction with the pathway TOR (Target Of Rapamycin) which is conserved across the living kingdoms of eukaryotes, and in this work studied for plants and fungi. The work included is extensive and represents an extremely conscientious effort logically to develop arguments for the importance of the discovery. However, although the work stands without question, as offered in this paper, some of the conclusions particularly those relating to the evolutionarily recent signalling role of the particular compound, 3-hydroxypropylglucosinolate (3OHPGSL), observed making this novel interaction, there are further considerations requiring more experimental work that I believe are necessary before such conclusions can be made with confidence. First of all it will be noted that I have run together the name of this compound, which is essential in describing the glucosinolates central to this publication. The particular glucosinolate is not a salt of the 3-hydroxypropyl ion nor is it an ester like molecule as in 3-hydroxypropyl acetate but is a glucosinolate substituted by 3-hydroxypropyl.

We apologize for having utilized an adaptation of the spelling to try and make the compounds more approachable for non-biochemical specialists. We have previously noticed that molecular/mechanistic audiences have trouble with the name when merged into a single word. We have changed all the spellings back.

Turning now to the evolutionarily related arguments and conclusions, these may well be correct but, in my view, do not stand without the following considerations and related new experimental work. Because the activity of 3OHPGSL is found beyond the plant order Brassicales then the authors are correct in eliminating the potential role of specific catabolitic enzymes acting on the glucosinolates in generating active species. 1) However, they have not proved satisfactorily that it is the intact 3OHPGSL molecule that has the observed activity. 2) If the study was extended to, for example 2-hydroxy-3-butenylglucosinolate then they would have a glucosinolate example more closely related structurally to 3OHPGSL. Also this would provide an example of a compound similar to 3OHPGSL in that it is potentially capable of undergoing related chemical changes which could take place beyond the Brassicales.

Please see our response above describing our new experiments that have been conducted per this request.

3) A further study, though less essential than 1 and 2, would be to investigate cyanogenic glycocides because these are considered to be the evolutionary ancestors of the glucosinolates. This would set in a more clear perspective the assertions regarding the recent evolution of the glucosinolates

We agree that the cyanogenic glucoside enzymes are the evolutionary precursors for the enzymes in glucosinolate production. However, as we have shown with the glucosinolates, it is not all glucosinolates that have this property but a limited structure. As cyanogenic glucosides do not have the sulfate, thioglucose or ability to make thio/isothiocyanates, we are unsure of a cyanogenic glucoside that would have similar structural properties or potential to the 3OHP GLS. The commercially available cyanogenic glucosides are structurally very different and as such we do not feel a comparison would be illuminating for this specific mechanism. It would however be very interesting to take all available cyanogenic glucosides and glucosinolates and feed them to plants to assess how many have potential signaling capacities and what traits they affect. We feel that this is beyond the scope of this manuscript however.

Reviewer #2:

This manuscript describes the intriguing finding that a specific secondary metabolite, 3-hydroxypropyl glucosinolate (3OHP GSL), can inhibit the universal eukaryotic TOR signaling complex, which the producing plant Arabidopsis thaliana appears to have adapted to use as a regulatory measure for balancing growth and development with the available energy and nutrients. The evidence for this use is strong, including identification of the target as TOR itself, and the authors have further investigated the downstream signaling pathway and effect, which appear to act on autophagy. However, it is less clear that this is a generalizable regulatory mechanism, which the authors suggest might be the case. Although softened to some extent with appropriate caveats within the Discussion, this is perhaps oversold in the Abstract.

We apologize for the confusion, we had not meant that the specific mechanism we identified 3OHP/TOR is generalizable but instead what may be generalizable is that secondary metabolites can have unrecognized regulatory properties. We quite agree that 3OHP/TOR is likely not generalizable in its specific context. We apologize for the misleading phrasing and have worked to make it clearer in the revised manuscript that it is the broader concept rather than this specific interaction that we believe is generalizable. This includes removing the idea that these would all be linked to optimization of energy usage. We realized after the reviewer’s comment that this was an over-extension of what we had intended.

One issue that is perhaps not fully discussed is the uncertain means by which A. thaliana has adapted to this effect. The generality of the effect of 3OHP GSL on TOR signaling in other, non-producing plants, as well as yeast, demonstrate that TOR itself is conserved. Moreover, given the similar phenotypic effect of 3OHP GSL on other plant species, it is not entirely clear how much adaptation has actually occurred versus selection for (or at least a lack of selection against) the (apparently) serendipitous TOR inhibition exhibited by this natural product. It would be helpful if the authors include some discussion of this point.

We agree that the conservation of the growth to 3OHP effect shows that TOR is conserved but we do feel that this helps to support that a conserved system like TOR is the system mediating this response to this compound (or a derived compound). We agree that it is not presently clear where selection/adaptation in Arabidopsis or any other plant may be in regards to responding to this serendipitous activity. To clarify we have boosted this section of the Discussion by commenting that a closely related Arabidopsis (*A. lyrata*) also has this compound suggesting that the responsible biosynthetic enzyme (AOP3) evolved in the last several million years. Further, in the revised section we now mention that we presently do not have evidence on what may be the benefit/detriment to the plant in having this 3OHP/TOR linkage. To obtain this knowledge will require identification of the putative receptor and conducting field trials to assess the consequences on the wild plant. We do have evidence of natural variation in Arabidopsis for the response to this compound and have noted this in the discussion but feel that including this data is beyond the focused scope of this manuscript.

As well as be a bit more realistic about the use of evolutionary young defense metabolites as regulators, which seems almost certain to require presumably serendipitous activity against already extant signaling pathways (although another possibility is that this activity is related to their defensive role, which would require proof of greater activity against the 'target' versus 'native producer' organism).

We completely agree that serendipity is likely a core requirement of this model. We have therefore worked to improve the text where we were attempting to make this point to enhance the clarity. We are very hesitant to propose any defensive role for this interaction as we have no evidence of the compound being exuded into the rhizosphere. As this is serendipity, there may not have been enough time to evolve a defensive role for this property.

It seems likely that the controls included application of the carrier solution, but this is not immediately evident in the Materials and methods description, and needs to be clarified.

This is correct and we have adjusted the Materials and methods to clarify the treatment plates were created by adding a concentrated sterile solution of 3OHPGSL in water to the agar post-sterilization to create media with the necessary concentration. The same methodology was used for the tests with the other GSLs tested. Control plates were created by adding the water to the control plates to replicate the process.